# Should Cognitive Differences Research Be Forbidden?

**Gerhard Meisenberg** 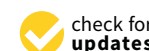

Ulster Institute of Social Research, 28 Haycroft Gardens, London NW103BN, UK; meisenberggerhard@gmail.com

**Abstract:** Some authors have proposed that research on cognitive differences, including differences between ethnic and racial groups, needs to be prevented because it produces true knowledge that is dangerous and socially undesirable. From a consequentialist perspective, this contribution investigates the usually unstated assumptions about harms and benefits behind these proposals. The conclusion is that intelligence differences provide powerful explanations of many important real-world phenomena, and that denying their causal role requires the promotion of alternative false beliefs. Acting on these false beliefs almost invariably prevents the effective management of societal problems while creating new ones. The proper questions to ask are not about the nature of the research and the results it is expected to produce, but about whether prevailing value systems can turn truthful knowledge about cognitive differences into benign outcomes, whatever the truth may be. These value systems are the proper focus of action. Therefore, the proposal to suppress knowledge about cognitive ability differences must be based on the argument that people in modern societies will apply such knowledge in malicious rather than beneficial ways, either because of universal limitations of human nature or because of specific features of modern societies.

**Keywords:** intelligence; individual differences; race differences; values; philosophy

## 1. Introduction

In a recent article, Janet Kourany asked whether knowledge about differences in cognitive ability (aka intelligence), and especially differences between categories of people such as genders and races, should be forbidden [1]. Kourany framed her argument as a conflict between the harms done by cognitive differences research and the freedom of scientists to pursue the research. Unlike some other scientists and philosophers, for example James Flynn [2], Kourany does not acknowledge benefits produced by this kind of research. However, the main weakness of her argument is that as evidence for the harms done by this research, she cites only a single study, which describes lower performance of women on a mathematics test in the stereotype threat paradigm [3]. In fact, the reproducibility of this effect is doubtful [4]. The black–white test score gap in the United States is also not explained by stereotype threat even assuming that acute stereotype threat effects are real [5]. In general, stereotype threat research is plagued by small and erratic effect sizes, and by publication bias [6].

In this contribution, I will not weigh possible harms against the "right" to pursue the research. Unless rights are understood purely instrumentally, the strength of two arguments cannot be compared when one is based on consequences while the other is based on rights. Doing so would be like comparing the weight of an object with the temperature of another object. I rather pursue the consequentialist approach of pitting the harms and benefits of the research against those done by preventing the research or actively suppressing the knowledge generated by the research.

The cognitive architecture that produces controversy about intelligence research is obvious to everyone who is even superficially acquainted with human psychology. As social primates, humans evolved cognitive specializations for functioning in dominance hierarchies. When confronted with the concept of intelligence, especially of "general" intelligence or *g*, humans interpret it as a marker of

dominance status [7]. It has been shown that individuals high on social dominance orientation tend to be more supportive of intelligence testing than those with egalitarian preferences [8]. It is plausible that the former self-select into intelligence research. It is equally predictable that those most engaged in controversies about intelligence research on either side tend to be unsavory characters that are focused on social dominance. Note that the proposed causal arrow points from preoccupation with social dominance to engagement with intelligence differences, rather than the reverse.

These automatisms are expected to be strongest in academics and public intellectuals because the appearance of high intelligence is an important determinant of career success in these professions. Therefore some of the academic acrimony about intelligence research may be unintelligible to those outside this subculture. Conversely, academics will have difficulty understanding the impact (or lack thereof) that knowledge about intelligence differences has on "real" people.

The aim of the present paper is to explain the controversy about intelligence research with reference to the psychology of its protagonists and the psychology of those whose worldviews are being affected by accumulating knowledge produced by intelligence research. Importantly, any harms and benefits of knowledge about cognitive ability differences depend on human values and their underlying psychology. The assumption is that moral values determine people's responses to the knowledge while being themselves largely unaffected by the knowledge. This permits two responses to "dangerous" knowledge: (1) treat human values as given (and as rather deficient), and ensure that knowledge never goes beyond the kinds that the value system can handle, as is implicit in most arguments against intelligence research; or (2) upgrade and refine the existing deficient value systems with the aim of enabling them to produce good outcomes for everyone based on the newly gained knowledge. Specifically, this paper examines evidence related to the following questions:

(1) What harms are expected to arise from knowledge about intelligence differences?

(2) More specifically, do moral values and cognitive abilities in modern societies make harmful consequences of true knowledge likely?

(3) Are these moral values and cognitive abilities malleable and responsive to educational interventions, and should such interventions be tried?

(4) Does knowledge about intelligence differences offer any benefits?

(5) Are alternative false explanations for socially important outcomes more beneficial than true explanations that are based on intelligence differences?

(6) What relationship between science and society do we desire, and how will it be affected by the suppression of research into cognitive ability differences?

First, however, we need to ask whether cognitive differences research can approach "objective knowledge" despite pervasive biases, and what the nature of these biases is.

## 2. The Question of Scientific Objectivity

Skepticism of science is justified because much of the research published in high-impact academic journals is irreproducible, in fields ranging from psychology to cancer research [9,10]. This includes many, even crucial and for the field central studies in psychology such as those about social priming [11]. However, when research produces results that challenge established worldviews or is expected to do so, a possibly unjustified distrust of the scientists and especially their motivations is a frequent response [12,13]. Therefore, I will briefly summarize the most important sources of bias:

(1) *Short-cuts*: Intense competition and career pressure induce scientists to engage in a plethora of questionable research practices [14]. Together with publication bias this is the most important source of false positive results in the empirical literature, but it is not specific to intelligence research.

(2) *Financial gain*: IQ tests are designed by test publishers to satisfy customer demand. For example, Matarazzo noted: "From the very beginning test developers of the best known intelligence scales (Binet, Terman, and Wechsler) took great care to counterbalance or eliminate from their final scale any items or subtests that empirically were found to result in a higher score for one sex over the other." [15]. This fiddling with cognitive tests creates a systematic bias towards gender equality

when the tests are used to study sex differences. However, its limitations are shown by the fact that despite enormous customer demand, test publishers could not produce a test that predicts important real-world outcomes while having no "adverse impact" on disadvantaged groups (poor people, Blacks, Hispanics . . . ). Whenever a test publisher claimed a markedly reduced adverse impact, users found the claim untrue or exaggerated [16,17].

(3) *Opportunism*: People are prone to shoot the messenger who brings them bad news. Therefore scientists, journal editors, and all those who make a living by debating and popularizing science have to cater to the preferences of their audiences, systematically promoting soothing falsehoods at the expense of grating truths. This is one reason for the marked disconnect between the beliefs of intelligence researchers and those of non-experts [18].

(4) *Wishful thinking*: Science-warping opportunism exploits widespread preferences among audiences. Passionate preferences about intelligence usually arise from one or more of four psychological needs: (a) The "conservative" preference for a hierarchical society in which social roles are assigned based on personal characteristics that are difficult to change (favoring intelligence research); (b) the "liberal" preference for an egalitarian society (opposing intelligence research); (c) the desire to improve oneself or the world, which would be foiled by claims that intelligence is both important and difficult to manipulate; and (d) the need to believe in a just world in which everyone is born with the same intellectual potential and the same opportunity for a good life [19]. The belief in Mother Nature's justice is an atheist theodicy: a substitute for the belief in divine justice through compensation in the afterlife. It predicts that intelligence research, and especially research on the genetics of intelligence, is more objectionable to atheists than to religious people.

(5) *Virtue signaling*: This form of opportunism is based on a shared understanding that people believe whatever makes them feel good. Nice people who want everyone to have the same opportunities in life believe that intelligence is either not quite real (e.g., "socially constructed"), unimportant, or the same for everyone; and nasty people believe that their chosen undesirables are condemned by bad genes and low intelligence. In consequence, virtue signaling demands visible opposition to intelligence research. Virtue signaling ought to be perceived as offensive by its targets because it implies that they are seen as morally deficient in being unable to acknowledge the existence of harsh realities that they find undesirable and in need of corrective action.

Are these biases sufficient to doom the enterprise of intelligence research? Nathan Cofnas argued that the science of cognitive differences is not self-correcting because the demand for true knowledge is outcompeted by the demand for false beliefs in the marketplace of ideas. Cofnas derived this conclusion from the pronouncements of prominent late 20th century intellectuals that show pervasive hostility to intelligence research and willful obstruction of science [20]. However, judgments about the ability of science to produce true knowledge should be based on the historic progression of the knowledge produced, not on the popularity of "alternative facts" promoted by career-conscious or socially concerned intellectuals.

The classical example is the heritability of intelligence. In 1969, Arthur Jensen proposed that the malleability of intelligence is seriously limited by its high heritability, based on results from educational interventions in the United States and the handful of heritability studies available at the time. He went on to suggest that race differences as well as individual differences are based on genetics although methods for testing this conjecture were not yet available [21]. Most responses were hostile, and books purporting to reject Jensen's conclusions were not only pop science blockbusters but were frequently cited in academic journals. For example, up to March 2019 Steven Jay Gould's 1981 polemic *The Mismeasure of Man* [22] achieved 11,139 citations according to Google Scholar, more than double the 5027 citations of Jensen's article.

Still, since the 1980s virtually all intelligence researchers had accepted the accumulating evidence showing high heritability of intelligence, and detailed knowledge from large twin studies is available today [23]. Rather than producing alternative facts for the benefit of their careers, the scientists produced what has become one of the most consistent bodies of knowledge in the behavioral sciences. Any

lingering doubts were dispelled by genome-wide association studies, the most recent of which identified 1271 chromosomal regions harboring common polymorphisms associated with intelligence [24]. The example shows that scientific knowledge is determined by supply as well as demand.

The argument we have to discuss in the following section is that cognitive differences research is dangerous precisely because it produces true knowledge.

## 3. Concerns about Harms of Knowledge, and the Refutation of Some of Them

Existing critiques of intelligence research offer few specifics about the harms that could accrue from knowledge about cognitive ability differences [1,25]. Therefore, I will briefly discuss the most plausible harms:

(1) *Lowered self-esteem*. Whenever two groups are compared, one will get a lower average score than the other. It can be argued that members of groups identified as low-scoring will lose self-esteem and develop serious pathologies as a result. This argument assumes that like academics, common people consider intelligence more important for their self-worth than other traits such as good looks, virtue, religiousness, criminal competence, or number of sex partners. Without this assumption, research into these other traits would be at least as objectionable as is intelligence research.

In addition, there is little evidence for negative effects of low self-esteem on important outcomes. Low self-esteem is statistically associated with a plethora of undesirable behaviors, personality traits, and life events, but the causal direction is more likely from experienced problems and underlying psychopathology to low self-esteem rather than the reverse. Therefore, concern about harms done by lowering people's self-esteem may be misplaced [26].

(2) *Self-fulfilling prophecy*. There is evidence that teacher expectations of individual students can affect changes in academic performance and cognitive test scores. The classical research paradigm involves giving teachers random information about the intellectual potential of individual children, with the effect that this information leads to actual changes in children's academic performance and cognitive test scores [27,28]. Expectancy effects appear to be real, and although they are of very small magnitude, they tend to be greatest in low-scoring children [29]. Therefore, even false claims of high ability for members of disadvantaged groups may make a small contribution to reducing performance gaps. Madon et al. (1997) concluded: "[P]rejudice and stereotypes may lead perceivers to hold negative expectations for members of stigmatized groups far more often than positive expectations . . . One way to increase the chance that members of stigmatized groups will benefit from self-fulfilling prophecies is by instituting policy changes that encourage perceivers (e.g., employers, teachers) to hold realistic, but high, expectations for targets." [29]. One policy change that could be instituted would be the enforced denial that intelligence differences exist. Expectancy effects are the strongest argument for tweaking or concealing evidence of low performance or low innate ability of "stigmatized groups", but the expected benefits are small because people are quite good at detecting ability differences that actually exist [30].

(3) *Promotion of discriminatory attitudes*. Just-world belief not only favors the rejection of "natural" (especially innate) intelligence differences, but also leads to the moralistic fallacy: The intuition that everything that is natural is also morally right and worthy of preservation. Therefore we can hypothesize that just-world belief can induce people to devalue those who are found to be "naturally" low in intelligence. Although empirical evidence for this is so far lacking, we have to note that together with expectancy effects, this is a major concern about the knowledge produced by intelligence research.

At a more applied level, intelligence is an explanatory construct for real-world outcomes including school success, political participation, health habits, and much more [31]. Is this causal attribution harmful by producing discriminatory attitudes? To assess this possibility, we have to acknowledge that people spontaneously form lay theories to explain the phenomena they observe in the world. The opposite of knowledge is not ignorance, but false belief. Rather than asking whether the attribution of an outcome—such as lower earnings of black than white Americans—to intelligence differences is harmful, we must ask whether it is more harmful than competing alternative explanations. Low earnings, for example, can be attributed to low intelligence or to laziness. By denying that low

intelligence is a cause, we invite people to attribute poverty to laziness instead. Low intelligence is also a risk factor for crime [32], with "free will" as the most popular, though non-scientific, alternative explanation. The belief that criminals are as intelligent as everyone else strengthens attributions of free will. In countries with functioning institutions and low corruption, free will belief favors punitive attitudes [33].

The examples show that denial of intelligence as a causal force can harm disadvantaged groups that have above-average rates of poverty and crime. Attributions of blame to external factors such as discrimination, parents, schools, politicians, or the capitalist system may be even more benign than intelligence attributions. These attributions need to be made when they are supported by evidence, but they are highly divisive when they are false and people cannot agree on who should be blamed.

(4) *Fatalism*. In his classical 1969 article, Arthur Jensen suggested that efforts at raising children's intelligence would meet serious limitations, based on two observations: (a) the transient nature of compensatory preschool education's effects on children's IQ, and (b) the evidence of high heritability for intelligence. With the benefit of hindsight, we now know both premises to be true—and the conclusion to be false. We know the conclusion to be false because average population IQ in the United States kept rising for at least four decades after Jensen wrote his article, at a rate of about three points per decade [34,35]. This Flynn effect shows that there was still potential for rising intelligence in the US population during the 1960s.

Does this mean that the educational interventions and heritability studies on which Jensen's conclusion was based should not have been done or their results should not have been published? There was a risk of bad outcomes, by discouraging efforts at raising children's intelligence. However, false conclusions are mass-produced in all research fields, to be used as working hypotheses by other researchers who will refute them. To prevent research findings and hypotheses that would be damaging if acted upon uncritically by policymakers, we would have to outlaw the entire social sciences. A more productive approach would be to foster critical thinking both within and outside of science.

(5) *Restriction of intellectual freedom*. Like religious and ideological convictions, beliefs about intelligence can evoke strong passions, making them important for some people's emotional well-being. Hence, the argument can be made that knowledge about intelligence ought to be banned because it harms people by restricting their freedom to choose their preferred non-scientific beliefs. The counterargument is that people ought to be free to believe in scientific as well as alternative facts, the way we allow them to believe in evolution rather than divine creation. Paul Feyerabend concluded so much: "The separation of state and church must be complemented by the separation of state and science, that most recent, most aggressive, and most dogmatic religious institution." [36].

## 4. Means and Ends

Proposals for restrictions of basic research assume an alternative between harmful knowledge and beneficial false beliefs. Preventing the generation or dissemination of harmful knowledge will ensure that everyone has beneficial false beliefs.

This conclusion is invalid. By merely preventing the generation or dissemination of harmful knowledge, we do not ensure the universality of beneficial false beliefs. We rather let people choose between harmful false beliefs and beneficial false beliefs. To achieve the desired outcome, the suppression of harmful knowledge must be complemented by the active creation and promotion of beneficial false beliefs. What distinguishes the fabrication and dissemination of false knowledge from the suppression of true knowledge is the difference between acts and omissions. Both are attempts at replacing disfavored beliefs by favored ones.

At this point we may note a rhetorical paradox faced by those advocating restrictions on basic scientific research. For example, any person proposing that research on genetic race differences in intelligence is harmful reveals that either she believes that large race differences exist but wants people to believe in their absence, or she believes that such differences do not exist but wants people to believe in them. Either way she reveals her deceptive intent. In addition, there is no certainty that the beliefs

of those who propose restrictions on intelligence research are true. If, for example, the expected large genetic race differences do not exist, stopping the research will only perpetuate an old prejudice.

Any restrictions of scientific knowledge have to be implemented by bureaucracies that are guided by formal rules. Rose emphasizes that many restrictions on scientific research already exist [25], and Kourany recommends institutional review boards (IRBs) as enforcers [1]. However, IRBs require binding guidelines with precisely defined lists of prohibited research. Otherwise each IRB's decisions would depend on idiosyncrasies, such as the balance between liberal and conservative IRB members. What is prohibited by one IRB would be allowed by another. These guidelines also must apply at least to an entire nation and they must not change too frequently—at least, not as frequently as federal support for Planned Parenthood changes in the United States. This calls for a major restructuring of political institutions in Western countries.

The practical challenges are illustrated by molecular genetic research on intelligence. Thousands of single-nucleotide polymorphisms (SNPs) have been published as being significantly associated with educational attainment or intelligence [24]. These SNPs had been in online databases such as 1000 Genomes and HapMap long before their relationship with intelligence became known. Everyone can navigate these sites to examine whether differences in allele frequencies can explain race differences in intelligence. Some published and unpublished studies have done exactly this [37–39]. In this case, information about population allele frequencies would have to be either deleted from all publicly accessible databases or falsified whenever a SNP is found to be associated with intelligence.

One option is to restrict access to sensitive knowledge to government agencies and policymakers, creating a distinction between classified science and publicly accessible science. However, the predictable result will be leaks by disgruntled employees. This will necessitate coercive measures to prevent science bloggers from analyzing, discussing, and disseminating the leaked information, and compliance by academics and journalists whose task is to disparage the illicit knowledge and its originators.

Another challenge is that science is international. Notable exceptions, such as anthropology under Hitler and genetics under Stalin, have proved short-lived and are not considered great successes by most historians of science. Today, different aspects of intelligence research are considered dangerous in different countries. Multicultural Western societies quarantine intelligence research on their ethnic, racial, and religious minorities, an aspect to which other countries are indifferent. Islamist governments may look askance at studies relating intelligence to religiosity, and may be reluctant to permit studies comparing the effects of faith-based versus science-based school curricula on children's intelligence. Preventing knowledge that has been generated in China, Russia, or India from reaching scientists and the public in Europe or the United States will require strict controls on scientific publishing and the internet.

## 5. The Dangers of Intelligence Research

In the following I will examine which specific knowledge about intelligence differences is sufficiently dangerous to merit prohibition, being aware that an outcome that is considered harmful by one person may be considered desirable by another.

(1) *Sex differences*. In Western academic culture, objections to knowledge about sex differences in intelligence are based on the belief that (1) the pervasive differences in gender roles that are observed world-wide are based on intelligence differences; (2) knowledge about intelligence differences, rather than their mere existence, perpetuates traditional gender roles; and (3) their denial promotes gender equality. I will focus on the defensibility of the first of these three assumptions.

Today, some experts (e.g., Richard Lynn) claim a very slight (3–4 points) male advantage in general intelligence during adulthood while others (e.g., James Flynn) deny it. Others again (e.g., Roberto Colom) deny sex differences in general intelligence but emphasize differences in specialized cognitive skills such as verbal and episodic memory, mental rotation, "emotional intelligence", and mechanical comprehension, some favoring males and others favoring females [40–42]. This contrasts with much

larger non-cognitive differences, for example a one standard deviation difference in vocational interests on the people–things dimension [43], and even larger differences when multivariate effect sizes of latent factors are determined [44]. Non-cognitive sex differences, including lower female risk taking and competitive spirit, are generally recognized as the main obstacles for attempts at educating women into traditionally male value systems and occupations [45]. There is also evidence that female relative to male happiness and life satisfaction tend to be high in Muslim countries and in countries with low female labor force participation [46]. This kind of knowledge, rather than knowledge about female intelligence, should be prohibited in societies whose elites disparage traditional female values and social roles!

(2) *IQ-based meritocracy*. This hypothesis is defined by Herrnstein's syllogism: (1) If differences in mental abilities are inherited, and (2) if success requires those abilities, and (3) if earnings and prestige depend on success, (4) then social standing will be based to some extent on inherited differences among people [47]. It has been claimed to apply to the United States [33], but not to Saudi Arabia and Sudan, which do not have a long history of educational sorting and where intellectual giftedness of children is only minimally related to parental socio-economic status [48,49]. The meritocracy hypothesis proposes that social stratification of intelligence, much of it transmitted genetically from parents to children, explains poverty and social pathologies afflicting those at the lower end of the IQ bell curve. There are three kinds of knowledge that support the hypothesis and that are candidates for prohibition, each of them already having strong scientific support: the heritability of intelligence [23]; the importance of intelligence for earnings, prestige, and social standing [50]; and social mobility as a consequence of familial transmission of intelligence genes [51].

If the meritocracy hypothesis is true, one likely result of its denial is an avalanche of false accusations, with blame for poverty and other bad outcomes attached to discrimination, a "culture of poverty", underfunded schools, incompetent parents, prejudiced employers, devious intellectuals and politicians, or the capitalist system. A second option employs concepts of free will, personal responsibility, and deservingness. These are the intellectual manifestations of cognitive routines for reciprocity: heuristics that we apply to control other people's behavior by positive and negative reinforcement [52]. Few people realize that "free will" is not an alternative to genetic and environmental causation, but the mechanism through which genes and environments control human behavior. Free will belief is associated with moralistic standards applied to self and others, just world belief, retributive punishment, and generally with "conservative" values [53,54]. When used as an alternative to genetically based ability differences, it leads to victim blaming: the belief that the poor do not deserve sympathy or assistance because their poverty is their own fault.

Revealingly, the "Bell Curve Wars" of the 1990s coincided with a virulent (and bipartisan) anti-welfare movement in the United States that was driven by a belief in the undeservingness of the poor [55]. In the United States, genetic causes of poverty and other social disadvantages, and especially the belief in genetically caused intelligence differences, had been repudiated by the intellectual elite since the 1960s. With innate ability differences out of the debate, intellectuals blamed the persistence of poverty and social pathologies on other intellectuals, while everyone else applied the deservingness heuristic to blame the victims—the "black welfare mother", in particular.

(3) *Race differences*. IQ differences between racial groups have been found in many countries [56]. The immediate concern in multi-racial countries is that knowledge of mean ability differences causes discrimination against individuals based on race rather than ability—the latter but not the former being considered acceptable. There is considerable a priori plausibility to this claim because race is assessed more easily than ability. Therefore vigilance to discrimination is required, although most empirical studies found no evidence of discrimination based on race when IQ and other predictors were controlled [57]. It is also questionable whether denial of true intelligence differences reduces discrimination because people are quite good at picking up regularities that actually exist in their world, especially regarding outcomes they care about [30].

The main concern is not about race differences in intelligence per se, but about possible genetic origins of these differences. The assumption is that human beings in general, or people in modern Western societies specifically, are incapable of respecting and valuing individuals when informed about low average innate ability of their race. Thus the suppression of knowledge is an alternative to the development of an ethos that values people irrespective of their personal intelligence or the average intelligence of their race (or social class, religion, occupation, etc.). Many thinkers have emphasized the separation between facts and morals:

> *"If someone defends racial discrimination on the grounds of genetic differences between races, it is more prudent to attack the logic of his argument than to accept the argument and deny any differences. The latter stance can leave one in an extremely awkward position if such a difference is subsequently shown to exist."*

<div align="right">John C. Loehlin et al., 1975 [58]</div>

> *"But it is a dangerous mistake to premise the moral equality of human beings on biological similarity because dissimilarity, once revealed, then becomes an argument for moral inequality."*

<div align="right">Anthony W. F. Edwards, 2003 [59]</div>

> *"Equality in spite of evident non-identity is a somewhat sophisticated concept and requires a moral stature of which many individuals seem to be incapable. They rather deny human variability and equate equality with identity. ... An ideology based on such obviously wrong premises can only lead to disaster. Its championship of human equality is based on a claim of identity. As soon as it is proved that the latter does not exist, the support of equality is likewise lost."*

<div align="right">Ernst Mayr, 1963 [60]</div>

Ernst Mayr spells out that ideology-driven (but not scientifically supported) denial of race differences reveals a lack of the "moral stature" required to grant (moral) equality despite (physical and mental) non-identity. Therefore, the important question is whether this lack of "moral stature" is limited to those who feel compelled to deny race differences for ideological reasons, or whether it is universal. If it is not universal, or is malleable by education, suppression of the knowledge should not be the preferred course of action.

The most popular and most divisive alternative to innate ability differences as an explanation for unequal outcomes is discrimination, conceptualized as "institutional racism", "symbolic racism", and similar constructs [61,62], sometimes bordering on conspiracy theories [63]. In the United States half a century after civil rights legislation and after intense efforts at ending discrimination, the belief in the continued power of racial discrimination implies the belief that discrimination is intractable. This leaves racial segregation as the only remedy. Conversely, genetically based race differences in intelligence can be used to argue that affirmative action policies should be maintained until undesirable intelligence differences can be eliminated by genetic engineering—unless we prefer maintaining or enhancing these differences instead.

(4) *Genes for intelligence*. How dangerous is knowledge about the genetic determinants of intelligence? In most situations, the polygenic scores that are used to predict intelligence will not raise fundamentally new ethical issues because they are not (much) more predictive than traditional demographics such as parents' education [64]. New ethical issues do arise when predictive genetic testing is used to select pre-implantation embryos for higher intelligence. In addition to embryo selection, knowledge of causal polymorphisms also allows for targeted genome editing to replace low-IQ alleles with high-IQ alleles and to repair genes that have been damaged by mutations [65].

Proponents of meritocracy advocate for a society in which environmental conditions are equalized for everyone such that success in life is determined by good genes. This genetically reinforced class society will be less stable when genetic as well as environmental differences have been minimized. Those who prefer stable meritocratic hierarchies should insist on the prohibition of knowledge about the molecular genetics of intelligence because it would lead to the demand for good genes for everyone.

For those unconvinced by the merits of meritocracy, the main concern is that a genetically based caste system will arise spontaneously because the more intelligent are more likely than the less intelligent to use genetic enhancements on their children, either because they are more intelligent or because they are more likely to be rich enough to afford them. Whether this concern is justified depends entirely on whether societies are willing to provide fair access to IQ-enhancing genes, the way they provide fair access to IQ-enhancing schools. Again, the pertinent question is not whether the knowledge is dangerous, but whether we have the ethical and intellectual stamina to use the knowledge for good ends.

(5) *History and evolution*. The intelligence level of populations can change rapidly in a single generation. Studies showing rising intelligence (Flynn effects) are welcome because they show that, in principle, we can raise intelligence by improving environmental conditions [31,32]. Even knowledge about declining intelligence is not usually considered objectionable although it questions the sustainability of modern societies and raises the specter of a downward spiral terminating in civilizational collapse [66].

However, knowledge of genetic IQ trends is a candidate for prohibition. A study based on ancient DNA reported that polygenic scores for education have increased in Europe since the Bronze Age [67]. If reproducible, this result can explain the advance of European civilization over the last three millennia. Western academics will most likely object that this knowledge is "scientific racism", especially if future research shows that polygenic scores remained unchanged or were declining in other world regions, such as the Middle East, while they were rising in Europe. Such results would be incompatible with "essentialist" beliefs in stable race differences and with the equally essentialist belief in zero race differences for intelligence-related genes. We cannot know which of these three alternatives will be best supported by future research, but suppressing the scientifically supported knowledge will promote both of the alternatives, not just one of them. Again, it is by no means assured that true knowledge is more harmful than the alternatives.

Present-day evolutionary change is another kind of dangerous knowledge. A genetic trend for declining intelligence in modern societies had been predicted throughout the last century based on negative correlations of fertility with intelligence and education [68], and has recently been confirmed genetically [69,70]. This questions the long-term sustainability of civilization as we know it.

What makes this knowledge objectionable is the traditional view that the prediction of disaster creates a moral obligation to prevent or mitigate the outcome. However, few people are motivated to promote the welfare of future generations. Thus the knowledge can be harmful by causing a bad conscience. In consequence, a principled argument for prohibition of this knowledge should be based on: (1) the factual claim that people do not care enough about future generations (and even their own children) to give them "good genes" but are still prone to develop bad feelings about it; and (2) the ethical claim that people living today have no obligation to promote the welfare of future generations.

## 6. Science and Society

The belief that reason, based on true knowledge, improves the human condition by overcoming superstition and prejudice was at the very heart of Enlightenment philosophy. Science was viewed as a vehicle to moral as well as material progress. Postmodern opposition to intelligence research is a reversal of Enlightenment philosophy in being based on the frequently unstated belief that science *creates* prejudice and bad behavior. Stated differently, members of present-day societies in general, or their ruling elites specifically, are assumed to be lacking the will or the ability to use knowledge about intelligence (or anything else) to achieve good ends.

The argument can take three forms. The first, based on genetic determinism, describes human beings as genetically programmed automata. Divisions between in-group and out-group, whatever way defined, are inevitable, and when informed about the low IQ of an out-group, people invariably use the knowledge to exploit or destroy this group, while high-IQ out-groups are targeted with

pre-emptive aggression. This argument, derived from evolutionary psychology, tends to deny the possibility of moral progress for individuals and societies.

Social determinism holds that existing power structures ensure that many people, especially among the political and social elites, are unwilling to use the knowledge for beneficial ends. They will instead monopolize the benefits for themselves to the detriment of everyone else, for example by giving their children high-IQ genes that they deny to others.

Both arguments are poor justifications for restrictions on basic research because members of the ruling elites will have to impose and enforce these restrictions. These individuals will themselves be flawed, with predictably bad outcomes. We cannot rely on criminals to protect us from crime. Epistocracy is more defensible than genetic or social determinism. It states that every society has a minority of individuals who possess the moral and intellectual qualities that are required to control everyone else's knowledge for beneficial ends: Plato's philosopher kings and their modern incarnation, the Chinese Communist Party [71]. Therefore any argument for the control of scientific knowledge must be based on the presumed existence of, or advocacy for, one or another version of the political system proposed for the Greek polis by Plato and adapted to a modern nation state by Xi Jinping. Only, it can be hard to distinguish philosopher kings from a corrupt, self-serving elite.

In the absence of philosopher kings, we need to examine the implicit social contract between science and society. This social contract is based on reciprocity: financial support for scientists in return for technological progress and economic growth produced by science [72]. This model is less straightforward in the basic social sciences, including intelligence research. What benefits, if any, do intelligence researchers have to offer?

Intelligence researchers emphasize the usefulness of their tests in getting children into the kinds of schools and adults into the kinds of jobs that suit their mental abilities. Philosophers in the Enlightenment tradition can add that knowledge of cognitive ability differences promotes ethical behavior, by understanding and accepting the limitations of those less well endowed by nature. Finally, cynics hold that society pays social science and humanities professors for creating comforting beliefs for public consumption and justifications for the existing order.

The intelligence researcher's version of the social contract calls for cost-benefit analyses in applications of the knowledge, but not restrictions on the creation or dissemination of knowledge. The Enlightenment philosopher's version holds that people are able to combine true knowledge about intelligence differences with universalist worldviews and compassionate moral values, but is silent on the proper course of action when rational assessment leads to the conclusion that people are lacking this ability. The cynic's version endorses the suppression of true knowledge in favor of "alternative facts" as long as it makes people happy, pleases those in power, and promotes the cynic's social standing and career.

All social contracts are based on trust. Therefore, excluding intelligence research from the social contract requires destroying trust in intelligence research by targeted defamation of intelligence researchers, both living and dead. This strategy is being pursued by ideology-driven groups up to hate groups today [73], and may become public policy in the future. While destroying trust in intelligence research is the desired outcome, it is less clear whether distrust of science can or should be prevented from spreading from intelligence research to other fields of inquiry. If we cannot trust intelligence researchers to tell the truth about sex differences and race differences, how can we believe what historians tell us about the Holocaust? How can we trust climate scientists who warn of global climate change, or biologists who tell us that genetically modified food is harmless? Although each of these fields has its own unique problems with public trust, the dismantling of intelligence research will promote a general erosion of trust in science. Whether this is desirable depends on whether or not we prefer a society that is based on science and reason to the available alternatives.

## 7. Conclusions

We can now answer the six questions asked at the end of the introduction:

1.  *What are the expected harms?* In modern Western societies, the main concern is that people will use information about intelligence differences to harm designated "disadvantaged" (i.e., low-IQ) groups.

2.  *Are bad consequences likely given prevailing moral values and cognitive limitations in modern societies?* The fact that the psychology of discrimination is taken for granted by opponents of intelligence research shows that it is indeed prevalent in this specific group. There is, however, no evidence that it is all-important society-wide.

3.  *Can and should we upgrade these moral values and cognitive abilities?* Promoting value systems that permit the beneficial use of *any* knowledge is proposed as a preferred alternative to the suppression of true knowledge produced by science—at least, we should try.

4.  *Does knowledge about intelligence differences offer any benefits?* This knowledge is required for constructive problem solving whenever a social problem is caused by cognitive limitations.

5.  *Are alternative false explanations more beneficial than true attributions to intelligence differences?* There is a very high risk that alternative false or incomplete explanations (e.g., discrimination) lead to blaming of victims or innocent others.

6.  *How does suppression of scientific knowledge fit into the political landscape?* It requires a totalitarian political system and destroys public trust in science.

On balance, the conclusion is that in any functioning human society, true knowledge is more likely than false beliefs to lead to constructive outcomes.

**Funding:** This research received no external funding.

**Conflicts of Interest:** The author declare no conflict of interest.

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
