# Peer review of "Should Cognitive Differences Research Be Forbidden?"

_psych, doi:10.3390/psych1010021_

Round 1

Reviewer 1 Report

In a sense, one might argue that papers such as this are not really useful any more because they are argue for generally accepted principles and truisms. However, in the era of post-truth, this is unfortunately not the case. We still need sound counter-arguments against censoring or self-censoring scientific knowledge. 

The case against research in differential psychology is one such case which is under debate for about a century. Provided that arguments against carrying this type of research still appear among several cycles, it is important to have systematic refutations in the sake of both truth, knowledge, and the best interests of those in need o this type of research. This paper presents several sound arguments against censoring scientific research in this field. The paper is well written and elaborate. Therefore, I propose that is accepted for publication.     

Author Response

The revised version is attached

Reviewer 2 Report

This article argues against restrictions on research on cognitive differences among racial, ethnic, gender, and other groups.  It reviews various arguments and evidence for and against and concludes with some recommendations.

I have several general concerns with the paper: 1) it is is too long, 2) it lacks clarity and focus, and 3) it is not well organized.  

The strength of the paper is it evaluation of the empirical evidence concerning the negative impacts of research on cognitive differences.  I recommend an extensive rewrite of the paper that focuses specifically on this issue.  The overall thesis would be that the evidence does not support the thesis that such research is likely to produce more harm than good.

Minor point: It is possible to critique rights-based arguments by appealing to consequences.  One can argue that a right should be restricted to because the benefits of restriction significantly outweigh the harms.  For example, one might argue that gun ownership rights should be restricted to prevent mass shootings, etc. For these arguments to work, one needs compelling evidence that the benefits of restriction would far outweigh the harms.  In the case of IQ research, opponents would need to produce compelling evidence concerning the harms of such research.    

Author Response

This is already the main part of the paper: general aspects in section 3, bad effects of the implementation process (i.e., need for a totalitarian state) in section 4, dangers of specific topics of intelligence research in section 5, and consequences for social trust in section 6. One common theme throughout is that the false beliefs that will have to be offered to people as a substitute for through. The abstract has been modified to emphasize this common theme. 

Rephrased to make clear that this applies to rights understood in the deontological sense but not to rights understood as means to an end

Reviewer 3 Report

Thanks for the interesting paper.

While I think that the subject and paper itself are very important I see problems

1. in the structure of the manuscript and

2. in the wording.

There is no congruence between headings (the structure of the manuscript) and the four final conclusions. There are much too many asides, sometime as awkward remarks or odd examples, that distract the readers’ attention (e.g., “penis size”, “intellectuals blamed the persistence of poverty and social pathologies on other intellectuals, while everyone else applied the deservingness heuristic to blame the victims—the “black welfare mother””). Or for instance: “in academics and public intellectuals because showcasing one’s intelligence, rather than doing useful work, is the main determinant of career success in these professions” – that is quite interesting but you do not bring any evidence for this claim, it is distracting, in this way and for your central argument a useless remark.

First of all, think about your main message, I think the main message are the four conclusions. Then reorganize the manuscript around the four conclusions, best use them as headings or use four questions leading to your four conclusions. Rewrite your paper around those four new headings. Delete all asides and incidental remarks.

In the abstract should be also put the four conclusions.

In these days (Easter) many people like to hear Bach’s St. Matthew's Passion. Here, there is one theme varying in many ways. There is one clear message. Your paper should also have one clear message.

The referencing system of Psych is very impractical (this is not the fault of the author but of the editors). Use APA standard!

James Flynn has published several papers about freedom of research etc. I think it would be useful to pick up his thread:

Flynn, J. (2007). Arthur Jensen and John Stuart Mill. Cato Unbound, November(26), www.cato-unbound.org/2007/11/23/james-r-flynn/arthur-jensen-and-john-stuart-mill.

Flynn, J. R. (2018). Academic freedom and race: You ought not to believe what you think may be true. Journal of Criminal Justice, 59, 127–131.

Flynn, J. R. (2018b). Reflections about intelligence over 40 years. Intelligence, 70, 73–83.

Also check whether they are important for your argument and manuscript:

Ceci, S. J. & Williams, W. M. (2009). Should scientists study race and IQ? Yes: The scientific truth must be pursued. Nature, 457, 788–789.

Nyborg, H. (2003). The sociology of psychometric and bio-behavioral sciences: A case study of destructive social reductionism and collective fraud in 20th century academia. In H. Nyborg (Ed.), The scientific study of general intelligence (pp. 441–502). Oxford: Pergamon.

Woodley of Menie, M. A., Dutton, E., Figueredo, A.-J., Carl, N., Debes, F., Hertler, S. et al. (2018). Communicating intelligence research: Media misrepresentation, the Gould Effect, and unexpected forces. Intelligence, 70, 84–87.

Regarding the second conclusion: “In most cases, true knowledge is more likely than false beliefs to lead to beneficial outcomes.” Where is evidence for this in the manuscript?

Author Response

I don’t think such congruence is essential as long as the important objectives are stated in the introduction and revisited in the conclusions. The second half of the introduction is now rewritten extensively to achieve this.

I tempered down some of these asides, but they serve a function of emphasizing the arcane and parochial nature of academic debates about intelligence research. Academics should be made aware that viewed from an outside perspective, their debates do indeed look hilarious.

These criticisms of the reviewer are extremely helpful. Yes, there should be one clear message, with some subheadings. In my rewrite I now state the main overarching theme in the introduction: “The aim of the present paper is to explain the controversy about intelligence research with reference to the psychology of its protagonists and the psychology of those whose worldviews are being affected by ongoing progress in intelligence research. Importantly, any harms and benefits of knowledge about cognitive ability differences depend in their entirety on human values and their underlying psychology. The assumption is that moral values determine people’s responses to the knowledge while being themselves largely unaffected by the knowledge. This permits two responses to “dangerous” knowledge: (1) Treat human values as given (and as rather deficient), and ensure that knowledge never goes beyond the kind the value system can handle, as is implicit in most arguments against intelligence research; or (2) upgrade and refine the existing deficient value systems with the aim of enabling them to produce good outcomes for everyone based on the newly gained knowledge.” This is followed by some specific questions that the paper examines, which are answered in the conclusions (now 6 of them, rather than the original 4).  

Thanks for these references. I found the Flynn (2018) reference especially useful.

Section 5 contains a good deal of evidence for specific topics in intelligence research. Admittedly, much of the evidence is indirect and not conclusive. However, this is the normal situation in science. An important contribution that I don’t see in any other discussion of the topic is that harms and benefits of true knowledge about intelligence differences should not be discussed in isolation, but in comparison with alternative false beliefs that are offered to explain important real-world phenomena or that people make up themselves. This is important because of the suspicion that intellectual maldevelopments in Western societies (e.g., ideological polarization) are the unintended side effects of well-intended efforts to protect people from dangerous knowledge.

Round 2

Reviewer 3 Report

Thanks for revising the paper.
It has improved a lot, however, certain suboptimal places could be improved:

It would be better for a response to the reviewers to add quotes of the original review.

Generally, I suggest for references to do it like Lynn, combine APA-style with the not helpful [number] style, see:
Lynn, R. (2019). Reflections on sixty-eight years of research on race and intelligence. Psych, 1(1), 1–9.

Lines 29 and 31: 2x However
Lines 44-46: “When confronted with the concept of intelligence, especially of “general” intelligence or g, humans interpret it as a marker of dominance status [7]. In consequence, individuals high on social dominance orientation tend to be more supportive of intelligence testing than those with egalitarian preferences [8].”
Probably, there are only small effects. So it is not possible, to explain a lot with it. It is too apodictically worded.
49: “obsessed” – is too vigorously worded.
52: “showcasing one’s intelligence is a main determinant of career success in these professions” – Evidence? Reference? More cautious wording.
59: “in their entirety”, too harsh.
72-76: Why are “intelligence differences” so important, more important are intelligence effects.
82: “much of the research published in high-impact academic journals is irreproducible”; better: many, even crucial and for the field central studies, e.g. [give two examples]
90ff.: There is no connection between financial gains and e.g., no sex differences in IQ tests.
103: “This is a major reason for” – too harsh.
108f.: “the “conservative” preference for a hierarchical social order in which every individual and every group occupies a fixed position (favoring intelligence research);” – too harsh.
119: “evil people” – too harsh.
149-150: “The argument we have to discuss in the following section is that cognitive differences research is dangerous precisely because it produces true knowledge.”
Very good, central message, repeat it in the abstract and if possible in a heading!
484: “by ideology-driven hate groups” -> by ideology-driven groups up to hate groups
493ff., conclusions: Repeat with very few words the question before each conclusion.
497-498: “The fact that the psychology of discrimination is taken for granted by opponents of intelligence research shows that it is indeed prevalent in some sections of society.” What? Because some take it for granted it is true?
Conclusion 3 is no answer for question 3.
